# Tendon-Derived Mesenchymal Stem Cells (TDSCs) as an In Vitro Model for Virological Studies in Wild Birds

**DOI:** 10.3390/v15071455

**Published:** 2023-06-27

**Authors:** José Rivas, Axel Dubois, Aude Blanquer, Mazarine Gérardy, Ute Ziegler, Martin H. Groschup, Luc Grobet, Mutien-Marie Garigliany

**Affiliations:** 1Fundamental and Applied Research for Animals & Health (FARAH), Laboratory of Pathology, Faculty of Veterinary Medicine, University of Liège, Sart Tilman B43, B-4000 Liège, Belgium; jfarivas@uliege.be (J.R.); aude.blanquer@uliege.be (A.B.); mazarine.gerardy@uliege.be (M.G.); 2Fundamental and Applied Research for Animals & Health (FARAH), Laboratory of Embryology, Faculty of Veterinary Medicine, University of Liège, Sart Tilman B43, B-4000 Liège, Belgium; axel.dubois@uliege.be (A.D.); lgrobet@uliege.be (L.G.); 3Institute for Novel and Emerging Infectious Diseases, Friedrich-Loeffler-Institut, Südufer 10, 17493 Greifswald-Insel Riems, Germany; ute.ziegler@fli.de (U.Z.); martin.groschup@fli.de (M.H.G.)

**Keywords:** cadaveric stem cells, in vitro model, wild bird, *Turdus merula*, Usutu virus

## Abstract

The use of wild animals in research is complicated due to the capture and housing conditions, as well as to legal aspects, making it difficult to develop in vivo and in vitro models for the study of pathologies that affect these species. Here we validate an in vitro model of tendon-derived mesenchymal cells (TDSC) from Eurasian blackbird (*Turdus merula*) cadaveric samples. Through the expression of surface markers and the ability to differentiate into multiple lineages, the nature of the cells was confirmed. We then evaluated Mesenchymal Stem Cells (MSCs) as an infection model for the Usutu Flavivirus. To this aim, blackbird TDSCs were compared to Vero E6 cells, commonly used in Flavivirus studies. Both cells showed permissiveness to USUV infection as confirmed by immunocytochemistry. Moreover, TDSCs exhibited replication kinetics similar to, although slightly lower than, Vero E6, confirming these cells as a pertinent study model for the study of the pathogenesis of USUV. In this work, we isolated and characterized tendon-derived mesenchymal stem cells, which represent an interesting and convenient in vitro model for the study of wildlife species in laboratories.

## 1. Introduction

Research projects exploring for instance a trait specific to a wild animal species can hardly be performed using conventional laboratory animals [1]. Working with wild animals in the laboratory presents complications related to their capture, housing, and legal restrictions due to their protection status, making their use as in vivo models more difficult [2,3]. The use of in vitro models can address this problem, although they have several limitations and do not always replicate in vivo scenarios [4,5]. Still, in vitro models present certain advantages as they are easy to implement, require less maintenance, are cost-effective, and provide a highly controlled environment, by the way removing the ethical issues owing to using live animals [4,6]. They can also be used in screening in pharmacology and toxicology; studies in genomics, proteomics and metabolomics; and studies of biomarkers in diseases for example [5]. However, the isolation of primary cells from animals usually requires fresh samples, which are hardly available for wild animals.

Over the last decade, several studies reported the successful isolation of Mesenchymal Stem Cells (MSCs) from cadaveric tissues, addressing the ethical problems due to the use of live animals and thus facilitating access to in vitro models [7]. MSCs are undifferentiated, self-renewing cells that form populations displaying multilineage differentiation potential [8]. These cells can be isolated from various tissues including bone marrow, subcutaneous adipose tissue, spinal cord, and tendon [7]. From the latter, tendon-derived stem cells (TDSCs) can be easily isolated from tendons obtained from biopsies or from cadavers between 48–72 h post-mortem by digestion with collagenase type I [9,10]. The cells present colony formation, rapid proliferation, and a high potential for multilineage differentiation, being an interesting source of MSCs [11,12]. Stem cells are known for their therapeutic role in tissue regeneration, especially in applications such as skeletal muscle tissue engineering, for which their use has been described successfully [13,14,15]. In recent years, they have been reported as options for cancer treatment [16] and also, to understand the pathogenesis of infections by viral agents such as severe acute respiratory syndrome-coronavirus-2 (SARS-CoV-2) [17] or Flaviviruses [18]

Usutu virus (USUV) is an emerging mosquito-borne virus belonging to the genus *Flavivirus*, family *Flaviviridae*, with a single-stranded RNA of positive polarity. The *Flavivirus* genus includes zoonotic arboviruses such as Zika virus (ZIKV), Dengue virus (DENV), Yellow Fever virus (YFV), and West Nile virus (WNV) among others, which are responsible for hemorrhagic diseases and/or viral encephalitis in humans and several animal species [19]. USUV is transmitted to a wide variety of avian hosts via mosquito bites, which may result in different clinical forms depending on the species infected, ranging from asymptomatic infections to severe neurological disease and death. Among the most susceptible species are members of the *Strigidae* family and many Passeriformes, including the Eurasian blackbird (*Turdus merula*) [20]. At the moment, no in vitro models have been described to study the pathogenesis of USUV in wild avian species susceptible to infection [21].

Skin is the first site of replication of mosquito-borne Flaviviruses [22]. Among the target cells are the dermal fibroblasts, which are the most abundant cells in the skin and fulfill sentinel and structural functions [22,23]. It has been shown in co-cultures during DENV infections that human dermal fibroblasts (HDFs) have a crosstalk via soluble factors with human dermal microvascular endothelial cells (HDMEC) and dendritic cells (DCs). This increases the secretion of cytokines that decrease DENV replication [24,25]. It also modulates the activation and maturation of DCs that are responsible for the activation of the adaptive immune response [24]. These observations suggest that the role of fibroblasts is fundamental for the early control of infection by Flaviviruses [23]. Isolation of fibroblasts from wild birds is complex due to the difficulty in accessing fresh carcasses, by contrast with MSCs which can be isolated several days post-mortem [7]. Both cell types share a mesenchymal origin, similar membrane receptors, and in vitro immune response mechanisms [26]. Therefore, MSCs can be an interesting surrogate for fibroblasts.

The aim of this work was to isolate and characterize TDSCs and to assess these cells as an in vitro model of viral infection. The cells were isolated from cadaveric blackbird tendon tissue at least two days after death. The nature of the cells was confirmed by the expression of specific markers and their ability to differentiate into other mesenchymal lineages was assessed. Then, the permissivity of these cells to the infection by USUV was compared with Vero E6 cells by immunocytochemistry and the kinetics of viral replication by RT-qPCR, with similar results between both cells. This validates TDSCs as the first in vitro model of USUV infection in susceptible wild avian species.

## 2. Materials and Methods

### 2.1. Isolation of TDSCs

Cells were isolated from a found dead female blackbird, which was kept refrigerated for 2 days before being dissected. Briefly, both legs were removed under aseptic conditions. They were immersed in 10% povidone-iodine for 1 min and rinsed 2 times with phosphate-buffered saline (PBS). Flexor tendons of the tibia-tarsal joint were removed and cleaned of residual muscle. Then TDSCs were isolated as previously described by Shikh Alsook et al., 2015 [9]. Briefly, the cells obtained were seeded at a density of 4 × 10^4^ cells/cm^2^ into 24-well plates previously coated with 0.1% Gelatin from porcine skin (Sigma-Aldrich, St. Louis, MI, USA). The cells were incubated at 37 °C with 500 µL of Dulbecco’s Modified Eagle’s Medium (DMEM) low glucose (1 g/L), with Sodium Pyruvate (Gibco, London, UK), supplemented with 10% heat-inactivated fetal bovine serum (FBS; Biowest, Nuaillé, France), 2% chicken serum (Gibco, UK), 1% Antibiotic-Antimycotic (Gibco, UK) and MEM Non-Essential Amino Acid Solution (Lonza, Morristown, NJ, USA), adapted from [27]. Half of the medium was changed every 3 days for each well. When cells reached 80–90% confluence, they were detached using TrypLE Express Enzyme (1×) (Gibco, UK). Then cells were harvested and placed in a 75 cm^2^ flask to amplify the cell population. When cells reached 80% confluence, they were harvested and stored in liquid nitrogen using a cryopreservation medium consisting of DMEM supplemented with 20% FBS and 10% dimethyl sulfoxide for cell culture (AppliChem, Darmstadt, Germany), using Nalgene Mr. Frosty™ Freezing Container (Thermo Scientific, Bremen, Germany).

### 2.2. Characterization of TDSCs

To confirm the identity of the isolated cells, primers were designed to amplify the mRNAs of the positive markers CD29, CD44, CD71, CD73, CD90, CD105, and the negative markers CD14, CD34, CD45, regularly used for the characterization of mesenchymal cells [28,29]. Due to the absence of validated PCR primers or gene information on useful markers in blackbirds in the literature, they had to be predicted. For this purpose, the desired mRNA sequences were mapped from raw data from a transcriptomic analysis in blackbirds [30] using as reference the mRNAs of Swainson’s thrush [31], a species genetically close to the blackbird [32]. All bioinformatic analyses were performed using Geneious 10.2.3 software (Biomatters, Auckland, New Zealand). From the predicted mRNA sequences primers were designed using Primer3 [33]. Endpoint RT-PCR was performed using the Luna Universal Probe One-Step RT-qPCR Kit (New England BioLabs, Ipswich, MA, USA). Total RNA extracted from passage 3 TDSCs using the TANBead Nucleic Acid Extraction Kit (Taiwan Advanced Nanotech, Taoyuan City, Taiwan) was used as a template. The amplification conditions were as follows: retrotranscription at 55 °C for 15 min; denaturation at 95 °C for 10 min; followed by target amplification for 45 cycles (95 °C for 30 s, 57 °C for 30 s, 72 °C for 60 s); final extension at 72 °C for 2 min.

### 2.3. Multilineage Differentiation of TDSCs

The multilineage differentiation capability of blackbird TDSCs was assessed using osteogenesis, adipogenesis, and chondrogenesis StemPro Differentiation kits (Gibco, London, UK). Briefly, for osteogenic and adipogenic differentiation, cells were seeded into 12-well plates at a density of 5 × 10^3^ cells/cm^2^ and 1 × 10^4^ cells/cm^2^, and incubated for 24 h before the culture medium was replaced by differentiation media. For chondrogenic differentiation, a micro mass culture method was employed. Droplets (5 µL) of a solution containing 1.5 × 10^7^ viable cells/mL were seeded into 12-well plates and incubated for 2 h under high humidity conditions before the addition of the differentiation medium. Differentiation media were changed every three days. Non-induced cells were cultured in a growth medium as a control. Osteogenic, adipogenic, and chondrogenic differentiation were assessed by Alizarin Red S staining (Sigma-Aldrich, St. Louis, MI, USA), Oil Red O staining (Sigma-Aldrich, USA), and Alcian Blue staining (Sigma-Aldrich, USA), respectively.

### 2.4. Permissivity Assays

The permissivity of blackbird TDSCs was compared with that of Vero E6 cells, a cell line commonly used as a reference in Flavivirus amplification due to their high permissivity [4]. Both cells were seeded on coverslips in 24-well plates 18 h before infection at a concentration of 0.1 × 10^6^ cells per well using DMEM with 2% FBS. Then the medium was removed, and the cells were infected with USU-BE-Seraing/2017, a Europe 3 lineage Usutu virus strain (Genbank: MK230892) [34] at an MOI of 10. After 2 h the inoculum was removed and replaced by DMEM with 2% FBS. After 18 h, the cells were fixed with Paraformaldehyde (PAF) 4% and permeabilized with 0.05% Triton diluted in PBS. Then the presence of viral particles was evidenced by immunocytochemical (ICC) staining using a rabbit polyclonal anti-USUV antibody U433 [35] and an anti-rabbit secondary antibody conjugated with a Horseradish Peroxidase (HRP)-labeled polymer (EnVision + System-HRP from Dako, Santa Clara, CA, USA).

### 2.5. Replication Kinetics

Blackbird MSCs and Vero E6 cells were seeded in 6-well plates at a concentration of 0.3 × 10^6^ cells per well using DMEM with 2% FBS. Cells were then infected in triplicate at a MOI of 1; 0.1; 0.01 or 0.001 under the same conditions as mentioned above. For each sample, 200 µL of supernatant were collected at 12, 24, 48, 72 hpi. Total RNA was extracted from the supernatant samples as described above. Viral RNA copy number was determined by absolute quantification by RT-qPCR, using primers described by [36] and Luna Universal Probe One-Step RT-qPCR Kit (New England BioLabs, USA). RT-qPCR conditions were: retrotranscription at 55 °C for 10 min; then initial denaturation at 95 °C for 1 min; followed by 45 amplification cycles (95 °C for 10 s, 48 °C for 20 s, 72 °C for 20 s). The viral RNA copy number was calculated using a standard curve, as described previously [37]. The logarithmic conversion was performed to normalize the distribution of the data revealed as non-parametric. The data were then analyzed using ANOVA implemented in Jamovi [38].

## 3. Results

### 3.1. Isolation of TDSCs

After 48 h of culture, the cells began to adhere to the culture plates and elongate. At 5 days, the cells gradually proliferated and presented a typical spindle-shaped fibroblastic morphology (Figure 1A). After two weeks in culture, the cells presented 90% confluency, so the cells were trypsinized, establishing the initial passage (P0). When subculturing the cells in a 75 cm^2^ flask, they exhibited a homogeneous fibroblast-like morphology and wave-shaped growth (Figure 1B). When the cells reached 80% confluence, they were harvested and resuspended in a cryopreservation medium to later be stored in liquid nitrogen. Subsequently, when the cells were thawed, they reached 90% confluence after 5 days of culture. The cells maintained the same characteristic morphology until passage 11, where they began to show signs of senescence, as described previously in fibroblast-like cells [39]. These included a decrease in the number of cells attached to the flask when seeded, a slower growth of the cells, which did not exceed 60% confluence, a reduction in the harvest density, changes in cell morphology, an increase in cell and nuclear size and vacuolized cytoplasm.

### 3.2. Molecular Characterization of TDSCs

From raw data of a Eurasian blackbird’s transcriptomic analysis [30] we deduced the mRNA sequences from the positive TDSCs markers CD29, CD44, CD71, CD73, CD90, CD105, and the negative markers CD14, CD34, and CD45. Nucleotide sequence data reported are available in the Third Party Annotation Section of the DDBJ/ENA/GenBank databases under the accession numbers TPA: BK064237-BK064246. From these mRNA sequences, we designed the primers presented in Table 1.

The isolated cells expressed the mRNA of all positive markers mentioned above (Figure 2). In the case of the negative markers, with the notable exception of CD45, none of them was amplified. All the amplicons obtained were confirmed by Sanger sequencing. 

### 3.3. Differentiation of TDSCs

To confirm the cell phenotype, the in vitro ability to differentiate into multiple lineages was assessed. TDSCs proved positive for differentiation into osteocytes, for which calcium deposits in the extracellular matrix were observed, as confirmed by Alizarin Red-S staining (Figure 3A). In the case of differentiation into adipocytes, the lipid droplets in the cytoplasm were identified using the Oil Red-O stain (Figure 3B). Finally, the differentiation into chondrocytes was evidenced by an extracellular proteoglycan-rich matrix, which was confirmed using Alcian Blue staining (Figure 3C).

### 3.4. Validation of TDSCs as a Model of USUV Infection

Cytopathic effects (CPE) were observed in both blackbird TDSCs and control (Vero E6) cells after the Usutu virus infection. CPE was evidenced by the appearance of retractile, round cells, followed by cell death and destruction of the cell monolayer after 18 h of infection (Figure 4B,D). Through ICC staining, the viral antigen signal was evidenced in the cytoplasm and membrane of the cells that remained in the well (Figure 4).

To compare the viral replication kinetics in both cell types, the USUV genome was quantified by RT-qPCR from the supernatant of blackbird TDSCs and Vero E6 with various multiplicities of infection. Both cells successfully amplified USUV, the largest differences were observed at an MOI of 0.01, starting at 24 hpi, and reaching the replication peak at 72 hpi (*p* < 0.001). Although TDSCs had slightly lower viral loads than Vero E6 cells, they were capable of efficiently replicating USUV (Figure 5).

## 4. Discussion

In this work, we successfully isolated TDSCs from Eurasian blackbird cadaveric tissues. In different species, including humans, MSCs have been isolated from different cadaveric tissues such as bone marrow, fatty subcutaneous tissue, skeletal muscle, spinal cord, and brain [7]. In the case of the tendon, TDSCs have been isolated from horse cadavers up to 72 h post-mortem [9]. It is postulated that the viability of the cells after the death of the animal is due to dormancy or long-term quiescence, which is a mechanism of resistance to stress that is an attribute of stem cells in adult tissue [9,40]. In addition, the post-mortem stress process where the cell suffers hypoxia, lack of nutrients, and tissue dehydration/rehydration could contribute to the selection of more robust and undifferentiated stem cells compared to the more differentiated cells from living donors [41].

The characterization of these cells was based on the Minimal Criteria for Defining Multipotent Mesenchymal Stromal Cells [8]. Since antibodies reacting with specific cell markers are not available for Eurasian blackbirds, the characterization was performed based on morphological characteristics, mRNA expression of key markers, and the ability to differentiate into multiple lineages. Blackbird TDSCs expressed the positive markers CD29, CD44, CD71, CD73, CD90, and CD105 and lacked the expression of the negative markers CD14 and CD34, as described in other MSCs of avian origin [12,28,29].

Unexpectedly, the cells expressed CD45, a marker typical of hematopoietic cells and used as a pan-leukocyte marker and whose expression is not expected for MSCs [8]. In adult MSCs, the expression of CD45 has however been described in muscle regeneration [42]. Furthermore, it was shown that MSCs derived from bone marrow that express CD45 preserve their differentiation potential in multilineages and fibroblast-like morphology, similar to MSCs that do not express this marker [43].

The multipotency of blackbird MSCs was confirmed by differentiation into three different lineages, which is the biological property that most uniquely characterizes MSCs [8]. In this work, we differentiated TDSCs from blackbirds in vitro under the action of specific induction factors into osteogenic, adipogenic, and chondrogenic lineages. The differentiation was assessed using specific staining procedures, i.e., Alizarin Red S staining, Oil Red O staining, and Alcian Blue staining, respectively, as validated in other MSCs models of avian origin [12,28,29,44].

The use of wild animals as in vivo models presents several limitations due to the complications in their capture, accommodation, high levels of distress, and alteration of the welfare and survival. This in turn can affect the reproducibility of the experiments, making it difficult to use in vivo models with wild animals [3]. In addition, wild animals are protected by European legislation, allowing in vivo experiments only in exceptional cases [45], complicating their use as experimental models. In vitro models, thus present an advantage by addressing the ethical issues associated with the use of wild animals in research [4]. Owing to their simplicity and the fact that they provide a highly controlled environment, in vitro models allow the study of key processes of viral pathogenesis, and represent a cost-effective method for the validation of antiviral drugs and other applications [5].

Therefore, TDSCs can be an interesting model for the study of the pathogenesis of viruses that affect wildlife, such as USUV. This Flavivirus emerged in Europe around 1996 in Italy causing mass mortalities in birds, particularly in species of the Passeriformes order, among which the Eurasian blackbird was the most affected [46]. The arrival of USUV generated a significant negative impact on the population of this bird. For instance, in Germany, the population decline was estimated at 15.7% compared to the areas not affected by USUV [47].

For the study of the pathogenesis of Flaviviruses, several in vitro models based on primary cell cultures have been described focusing on the main organs affected by viral infection [4]. Those include peripheral blood mononuclear cells (PBMC), for the study of the immune response [48,49,50], and cells of the central nervous system for the study of neuropathogenesis [51,52].

Another important organ involved in the pathogenesis of Flaviviruses is the skin, this organ is the main route of infection and the first replication site from where the virus spreads to the rest of the organism [23]. Several in vitro models have been developed from this organ, mainly keratinocytes, melanocytes, and dermal fibroblasts [23]. The latter is one of the most abundant cells of the skin that fulfills a structural and sentinel function [25]. It has been shown that DENV-infected human dermal fibroblasts in co-culture with human dermal microvascular endothelial cells (HDMEC) show a crosstalk through soluble factors that increase IFNb secretion, thus decreasing DENV replication [25]. In addition to facilitating leukocyte migration through the HDMEC monolayer, this suggests that it might help in the early control of DENV at the site of infection [25]. Another important function of fibroblasts is the activation of dendritic cells (DCs) of the skin, which are the main link between innate and adaptive immune responses [53]. This activation results in the maturation of DCs and the subsequent activation of T cells [53]. In vitro, it has been observed that soluble factors of DC and fibroblasts infected by DENV modulate the activation and maturation of these cells, promoting the control of the infection and the activation of the adaptive immune response [24]. This makes fibroblasts a key cell in the early control of Flavivirus infection and a potential key player in the differences of susceptibility/resistance to the infection observed in vivo.

Currently, no in vitro models have been described in susceptible wild avian species to study USUV. The replication of USUV has however, been described in fibroblast obtained from clinically resistant species, i.e., chicken (*Gallus gallus domesticus*) and goose (*Anser anser f. domestica*) [21]. These species have been confirmed experimentally to be highly resistant to USUV infection [54,55]. Due to the in vitro characteristics of TDSCs, they can be a surrogate for fibroblasts since they have the same mesenchymal origin, morphology, and expression pattern, including during immune reactions [26]. Additionally, stem cells have previously been used for in vitro studies of the pathogenesis and immune response to Flavivirus infections, including ZIKV [18,56,57], DENV, YFV, and WNV [18].

In order to validate blackbird TDSCs as in vitro models to study the pathogenesis of USUV, these cells were compared with Vero E6, considered as a reference for the culture of Flaviviruses [21]. To this aim, we compared the permissivity of both cell types to USUV infection. The USU-BE-Seraing/2017 strain was chosen since in previous studies it presented the highest replication in primary cell models [58]. By immunocytochemistry a large number of viral antigens was observed in the cytoplasm of the cells, thus confirming that blackbird TDSCs are permissive to USUV. When comparing replication kinetics, blackbird TDSCs exhibited similar, albeit slightly less efficient, replicative behavior as Vero E6 cells. Although TDSCs have lower viral loads, these cells were able to efficiently replicate USUV. This replication difference may be due to the fact that Vero E6 cells do not express IFNa and IFNb [59]. In addition, it could be the consequence of the fact that the viral stock has been passaged 7 times in Vero E6 cells after isolation from a wild bird. Finally, we cannot exclude that different results might have been obtained if the intracellular (instead of extracellular) viral RNA concentrations had been compared between both cell types.

## 5. Conclusions

In this work, TDSCs were isolated from Eurasian blackbirds. This is a promising in vitro tool for research on wild birds, which due to protection regulations and complex capture and housing are difficult to study in the laboratory. Here we validated TDSCs as the first in vitro model to study the pathogenesis of the USUV virus in susceptible wild avian species. Further studies are needed to determine if TDSCs are a good model for isolating viruses freshly collected from birds and if this in vitro model replicates the susceptible and resistant phenotypes observed after USUV infection in avian species in vivo.

## Figures and Tables

**Figure 1 viruses-15-01455-f001:**
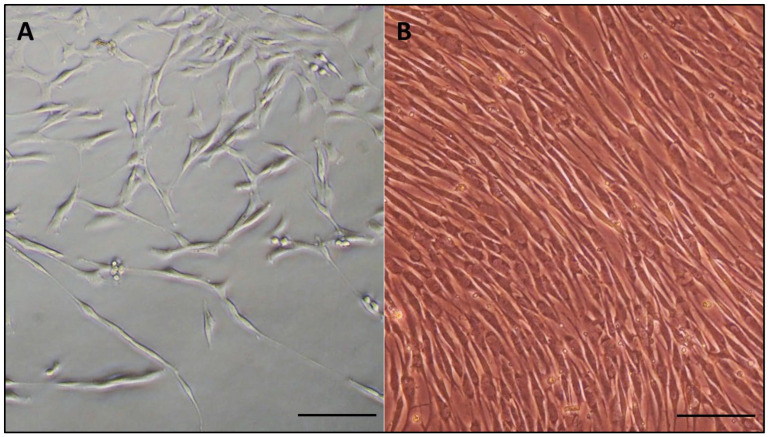
Primary culture morphology of blackbird TDSCs. (**A**) Typical spindle cells after 5 days of culture in 6-well plate, magnification 200×. (**B**) Confluent cells in subculture, passage 4 in 175 cm^2^ flask, magnification 200×.

**Figure 2 viruses-15-01455-f002:**
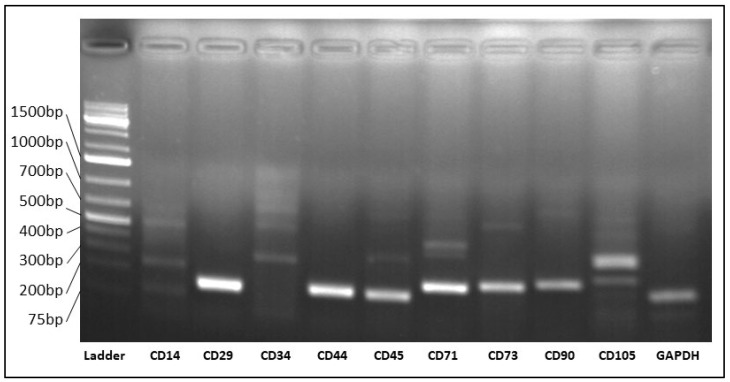
Expression of surface markers of TDSCs detected by end-point RT-PCR. Passage 3 TDSCs were positive for the expression of CD29, CD44, CD45, CD71, CD73, CD90, CD105 and negative for CD14 and CD34. GAPDH served as control.

**Figure 3 viruses-15-01455-f003:**
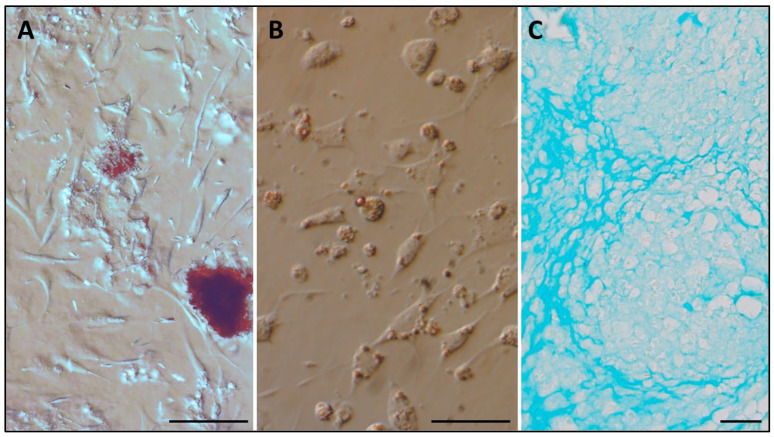
Cytochemical staining of blackbird TDSCs, after differentiation. (**A**) Alizarin Red-S staining. (**B**) Oil Red-O stain. (**C**) Alcian Blue staining. Scale bar = 50 µL.

**Figure 4 viruses-15-01455-f004:**
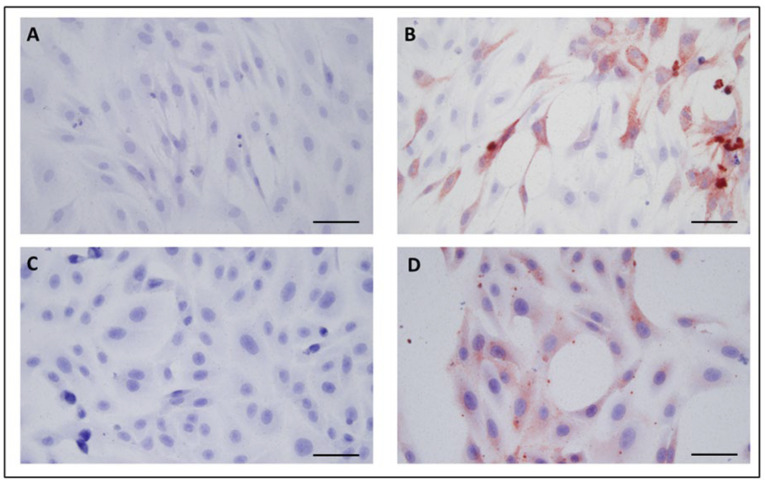
Immunocytochemical staining of USUV antigens performed on: (**A**) uninfected blackbird TDSCs; (**B**), USUV-infected blackbird TDSCs; (**C**), uninfected Vero E6 cells; (**D**), Vero E6 cells infected with USUV. MOI: 10; 18 hpi. Scale bar = 50 µL.

**Figure 5 viruses-15-01455-f005:**
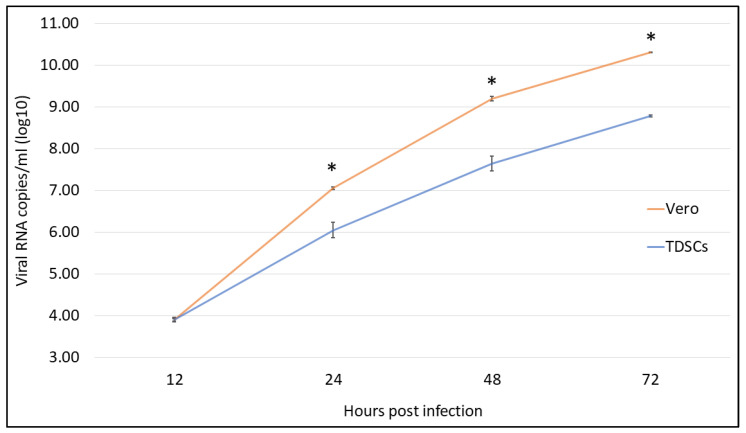
Viral RNA loads in the supernatants of Vero E6 cells (orange line) and blackbird TDSCs (blue line) infected with USUV at a MOI of 0.01. Error bars represent Standard Deviation, statistical differences were observed from 24 hpi (* *p* < 0.001, ANOVA test).

**Table 1 viruses-15-01455-t001:** Primer sequences used in the RT-PCR characterization of blackbird TDSCs.

mRNA	Accession Number	Primer	Primer Sequence	Amplicon Size (bp)
CD29	BK064246	CD29F	CATTCCCATTGTAGCCGGTG	151
		CD29R	TTCACCCGTATCCCACTTGG	
CD44	BK064237	CD44F	CCTTCTGGGTGCTGACAAAC	158
		CD44R	ATTTCCCCTGGTGTGGATCA	
CD71	BK064244	CD71F	AGATGACTCCTACTGCGTCG	200
		CD71R	GGCAGCGTTCTCATCTTCAG	
CD73	BK064243	CD73F	CCCATTGATGAGCAGAGCAC	211
		CD73R	CTGGGGCTTTGGAGAGATCA	
CD90	BK064242	CD90F	TCTCCGAGAACATCTACCGC	221
		CD90R	CCACGAGGTGTTCTGGATCA	
CD105	BK064241	CD105F	GCTGACTTCAAGGCACAACA	245
		CD105R	ATGGTGTAGGTGAAGCGGAA	
CD14	BK064239	CD14F	GTCGCCAGCTCAGTACCA	224
		CD14R	GGACACCAAGCACAGGGA	
CD34	BK064238	CD34F	GGCAGGAATTTGGGTGTGAG	233
		CD34R	TCATGTCCCTGCTCATCCTG	
CD45	BK064245	CD45F	TGACACCATTGCCAGTACCT	156
		CD45R	GTTTTCTCTGGCTGTGGTGG	
GAPDH	BK064240	GAPDH_F	TCTCTGTTGTGGACCTGACC	169
		GAPDH_R	TCAAAGGTGGAGGAATGGCT	

## Data Availability

Not applicable.

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
