# Peer review of "Tendon-Derived Mesenchymal Stem Cells (TDSCs) as an In Vitro Model for Virological Studies in Wild Birds"

_viruses, 2023, doi:10.3390/v15071455_

Round 1

Reviewer 1 Report

- In the summary, the sentence present in lines 24 to 26 was not very clear to my understanding, appearing to lack elements that improve its understanding. I suggest reviewing.

- The objectives of the study are not written in a clear and direct way, with notes that refer to the methodology topic. I suggest reviewing.

- I do not think that the information contained in lines 329 to 331 are consistent with the conclusions of the work. In my opinion, this information would be better suited in the introduction, reinforcing that TDSC cells have proven application in biomedical and pharmaceutical research. Thus, I suggest reviewing.

Author Response

Reviewer #1:

Comment 1: In the summary, the sentence present in lines 24 to 26 was not very clear to my understanding, appearing to lack elements that improve its understanding. I suggest reviewing.

Response comment 1:  We thank the reviewer for his/her suggestion. To improve understanding we replaced the sentence: “Tendon derived mesenchymal stem cells thus represent an interesting and convenient in vitro model for the study of wildlife species.” by the following: “In this work we isolated and characterized Tendon-derived mesenchymal stem cells, which represent an interesting and convenient in vitro model for the study of wildlife species in laboratories.

Comment 2: The objectives of the study are not written in a clear and direct way, with notes that refer to the methodology topic. I suggest reviewing.

Response comment 2: We thank the reviewer for his/her constructive comment. To clarify the objectives of this study, we suggest the following changes (lines 83-90): “The aim of this work was to isolate and characterize TDSCs and to assess these cells as an in vitro model of viral infection. The cells were isolated from cadaveric blackbird tendon tissue from at least two days after death. The nature of the cells was confirmed by expression of specific markers and their ability to differentiate into other mesenchymal lineages was assessed. Then, the permissivity of these cells to the infection by USUV was compared with Vero E6 cells by immunocytochemistry and the kinetics of viral replication by RT-qPCR, with similar results between both cells. This validates TDSCs as the first in vitro model of USUV infection in susceptible wild avian species.

Comment 3: I do not think that the information contained in lines 329 to 331 are consistent with the conclusions of the work. In my opinion, this information would be better suited in the introduction, reinforcing that TDSC cells have proven application in biomedical and pharmaceutical research. Thus, I suggest reviewing.

Response comment 3: We appreciate the comment of the reviewer. We moved the sentence: “In addition, such cells have been proposed for other applications such as skeletal muscle tissue engineering, for which their use has been described successfully [10,61]. They can also be used in screening in pharmacology and toxicology; studies in genomics, proteomics and metabolomics; and study of biomarkers in diseases for example [5].” to the introduction, with adaptations (lines 39-41 and 52-54).

Reviewer 2 Report

Rivas et al describe the production of TDSC cells and replication of Usutu Virus in these cells and compare to Vero E6 cells. The manuscript is well written but the images should be improved. 

Comments:

MSCs is not defined in the abstract

Line 46: please clarify / expand "isolated from digestion", as in the enzymes utilised or or variation within the 3 days stated.

Figure 1: Could the sample magnification be the same for both images and the phase contrast be adjusted, the cell morphology is not clear.

Figure 2: Please label the ladder 

Figure 3: Please use a better phase contrast, it is particularly difficult to identify the Oil Red-O staining and cell morphology in the Alcian Blue stain.

Line 212: Please check consistency of Vero E6 throughout the manuscript, in some places only referred to Vero.

Line 256: tree should be three

Line 323: The discussion could also comment on the PCR target, from the reference the target RT-qPCR is NSP-1, this may be expressed differently between the two cell lines and therefore contribute to the observed differences in viral loads observed in figure 5. From Figure 4 it appears there is more larger aggregates of viral staining within the TDSC cells than Vero E6, did the authors perform RT-qPCR on cell lysates to establish if more virus was retained within the cells i.e. a different viral distribution between the cell lines? 

Author Response

Reviewer #2:

Rivas et al describe the production of TDSC cells and replication of Usutu Virus in these cells and compare to Vero E6 cells. The manuscript is well written but the images should be improved.

Response: We thank the reviewer for his/her appreciation of our manuscript.

Comment 1: MSCs is not defined in the abstract

Response comment 1: Indeed. The Abstract was corrected (line 20): “Mesenchymal Stem Cells (MSCs)

Comment 2: Line 46: please clarify / expand "isolated from digestion", as in the enzymes utilised or variation within the 3 days stated

Response comment 2: We appreciate the comment of the reviewer. The authors of reference 9 (the only article reporting the isolation of TDSCs from cadavers) mentionned that "Four cadaveric forelimbs, received within 48–72 hours post-mortem (from horses 18–20 years old) were used in this study"; biopsies were used in the reference 10.

To make things clearer, we thus corrected our manuscript as follows (lines 48 – 50): “From the latter, tendon-derived stem cells (TDSCs) can be easily isolated from tendons obtained from biopsies or from cadavers between 48 - 72 hours post-mortem by digestion with collagenase type I [9,10].

Comment 3: Figure 1: Could the sample magnification be the same for both images and the phase contrast be adjusted, the cell morphology is not clear.

Response comment 3: We thank the reviewer for this constructive remark. The Figures 1A and B were replaced accordingly (same magnification and adjustment of the phase contrast).

Comment 4: Figure 2: Please label the ladder

Response comment 4: We thank the reviewer for this observation. The ladder was labeled and included in the Figure 2.

Comment 5: Figure 3: Please use a better phase contrast, it is particularly difficult to identify the Oil Red-O staining and cell morphology in the Alcian Blue stain.

Response comment 5: We agree with the reviewer. The Figure 3 was corrected accordingly (higher magnification and better contrast).

Comment 6: Line 212: Please check consistency of Vero E6 throughout the manuscript, in some places only referred to Vero.

Response comment 6: We thank the reviewer for the careful reading of the manuscript. We rigorously revised the manuscript and replaced “Vero” by “Vero E6”, wherever needed.

Comment 7: Line 256: tree should be three

Response comment 7: Indeed. Corrected (line 270).

Comment 8: Line 323: The discussion could also comment on the PCR target, from the reference the target RT-qPCR is NSP-1, this may be expressed differently between the two cell lines and therefore contribute to the observed differences in viral loads observed in figure 5. From Figure 4 it appears there is more larger aggregates of viral staining within the TDSC cells than Vero E6, did the authors perform RT-qPCR on cell lysates to establish if more virus was retained within the cells i.e. a different viral distribution between the cell lines?

Response comment 8: We appreciate both reviewer’s comments. Since USUV genome is a single strand of (positive) RNA containing a single open reading frame (ORF), the specific region of the genome targeted by the RT-qPCR does not affect the number of copies detected. This number of viral RNA copies is thus expected to be correlated with the level of replication of the genome, regardless of the cell type.

Regarding the second question, we did not assess the amount of viral RNA in cell lysates (only in the supernatant). The aim of the cytochemical assay was to qualitatively evaluate the permissivity of Blackbird TDSCs in comparison with that of the “gold-standard” Vero cells. We believe that it would be risky to make quantitative comparison based on a single high-magnification picture. We did not had such an impression on the whole slide microscopic evaluation. For this reason, we used RT-qPCR for unbiased quantitative comparison. There remains that we agree with the Reviewer that the amounts of intracellular viral RNA in both cell types might differ from what we measured extracellularly.

Due to the high relevance of this remark, we included the following sentence to the revised version of the manuscript (Lines 337 – 340): “Finally, we cannot exclude that different results might have been obtained if the intracellular (instead of extracellular) viral RNA concentrations had been compared be-tween both cell types. ”

Reviewer 3 Report

The protocol developed in this study should overcome the difficulties of wildlife research and is very interesting. When aiming to elucidate the mechanism of infection of natural hosts by pathogens (especially viruses that require host cells for their propagation), there have been a strong demand for the development of analysis methods using cells derived from actual host animals. The authors isolated Tendon-derived mesenchymal stem cells (TDSCs) from wild bird carcass samples and successfully infected and propagated TDSCs with a flavivirus, Usutu virus. The results presented here should provide an interesting and convenient in vitro study model for researchers of viruses carried by wildlife species.

[Major points]

Nothing

[Minor points]

1.      Line 156: Delete “in” and insert “previously”.

2.      Figure 1: If it is possible, enlarge figures to see the cell morphology more clearly.

3.      Table 1: If the authors got the accession numbers of the genes, describe in the table.

4.      References #60 and 61 are missing.

[Questions]

If possible, include the following points?

1.      How many times can TDSCs be subcultured?

2.      Do TDSCs undergo morphological and genetic mutations during passaging?

3.      Can TDSCs be cryopreserved?

4.      What is the growth efficiency of the virus when TDSCs are inoculated with virus freshly collected from birds?

Author Response

Reviewer #3:

The protocol developed in this study should overcome the difficulties of wildlife research and is very interesting. When aiming to elucidate the mechanism of infection of natural hosts by pathogens (especially viruses that require host cells for their propagation), there have been a strong demand for the development of analysis methods using cells derived from actual host animals. The authors isolated Tendon-derived mesenchymal stem cells (TDSCs) from wild bird carcass samples and successfully infected and propagated TDSCs with a flavivirus, Usutu virus. The results presented here should provide an interesting and convenient in vitro study model for researchers of viruses carried by wildlife species.

Response: Many thanks to the reviewer for the appreciation of our work and the positive comments on our manuscript.

Comment 1: Line 156: Delete “in” and insert “previously”.

Response comment 1: : We thank the reviewer for this suggestion. The manuscript was corrected accordingly (line 164 in the corrected version of the manuscript).  

Comment 2: Figure 1: If it is possible, enlarge figures to see the cell morphology more clearly.

Response comment 2: We thank the reviewer for his/her observation. The Figure 1 was replaced by one with greater magnification that allows better observation of the cell morphology.

Comment 3: Table 1: If the authors got the accession numbers of the genes, describe in the table.

Response comment 3: Unfortunately we do not have the accession numbers yet, the sequences are still under review at NCBI. As soon as we have the accession numbers they will be included in the manuscript.

Comment 4: References #60 and 61 are missing.

Response comment 4: Indeed. We revised the manuscript and corrected the missing references.

Comment 5: How many times can TDSCs be subcultured?

Response comment 5: Many thanks to the reviewer for this interesting question. In our hands, the cells could be successfully subcultured up to passage 11, where they began to show signs of senescence. Following the reviewer’s comment, we found pertinent to add this information in the revised version of the manuscript (lines 179-181): “The cells maintained the same characteristic morphology until passage 11, where they began to show signs of senescence, as described previously in fibroblast-like cells [39].”

Reference 39 was included in the manuscript: “Cristofalo, V.J.; Pignolo, R.J. Replicative Senescence of Human Fibroblast-Like Cells in Culture; 1993; Vol. 73.”

Comment 6: Do TDSCs undergo morphological and genetic mutations during passaging?

Response comment 6: We thanks to the reviewer for this question. Unfortunately genetic mutations were not assessed during the passages of these cells. On the contrary, morphological changes were observed at passage 11, likely associated with cell senescence, as mentioned in the response to comment 5. The details are specified in the following sentences that were included in the manuscript: “. The cells maintained the same characteristic morphology until passage 11, where they began to show signs of senescence, as described previously in fibroblast-like cells [39]. These included a decrease in the number of cells attached to the flask when seeded, a slower growth of the cells, which did not exceed 60% confluence, a reduction in the harvest density, changes in cell morphology, an increase in cell and nuclear size and vacuolized cytoplasm (data not shown).” lines 179 – 184.

Comment 7: Can TDSCs be cryopreserved?

Response comment 7: Indeed, TDSCs can be successfully cryopreserved. We included more details on the procedure in the manuscript, in the “materials and methods” section: “When cells reached 80% confluence, they were harvested and stored in liquid nitrogen using cryopreservation medium consisting of DMEM supplemented with 20% FBS and 10% dimethyl sulfoxide for cell culture (AppliChem, Germany), using Nalgene Mr. Frosty™ Freezing Container (Thermo Scientific, Germany).” lines 107 – 110. And in the “results” section: “When the cells reached 80% confluence, they were harvested and resuspended in cryopreservation medium to later be stored in liquid nitrogen.” lines 176 – 178.

Comment 8: What is the growth efficiency of the virus when TDSCs are inoculated with virus freshly collected from birds?

Response comment 8: We thank to the reviewer for this very interesting question. Unfortunately, we did not have the opportunity to isolate fresh virus from birds on TDSCs yet. We hope to address this question in the future. Due to the relevance of this question was added the following sentence to the manuscript: “Further studies are needed to determine if TDSCs are a good model for isolating viruses freshly collected from birds and if this in vitro model replicates the susceptible and resistant phenotypes observed after USUV infection in avian species in vivo.” lines 346 – 348.